

# Characterization and clinical outcomes of outpatients with subacute or chronic post COVID-19 cough: a real-world study

Chun Yao, Dongliang Cheng, Wenhong Yang, Yun Guo and Tong Zhou

Department of Respiratory and Critical Care Medicine, the Second Affiliated Hospital of Soochow University, Suzhou, Jiangsu, China

## ABSTRACT

**Background:** Limited research exists on the features and management of post-COVID cough. This real-world study investigates outpatients with subacute or chronic post-COVID cough, aiming to delineate characteristics and regimen responses within the population.

**Method:** We enrolled eligible patients from our outpatient unit between August 2023 and February 2024. Comprehensive clinical data, prescriptions, and patient-reported cough severity were collected during the primary visit and subsequent follow-ups.

**Result:** A total of 141 patients, aged: 42 ± 14 years old, were included, with 70% being female. The median cough duration was 8 weeks (interquartile range 4–12 weeks). Sixty percent presented with a dry cough, while the rest had coughs with phlegm. Over half reported abnormal laryngeal sensations (54%). Twenty-one percent coughed during the day, while 32% coughed constantly, and 48% experienced nocturnal episodes. Compound methoxyphenamine capsules were the most prescribed, but our study found ICS/LABA to be the most effective, followed by compound methoxyphenamine capsules, montelukast, and Chinese patent drugs.

**Conclusion:** Females exhibit a higher prevalence of post-COVID cough, and our study recommends ICS/LABA as the preferred treatment. These findings warrant validation through larger, prospectively designed studies.

## BACKGROUND

COVID-19 is a disease caused by a novel strain of coronavirus first identified in late 2019 (*World Health Organization (WHO), 2024a*). It is a respiratory infectious disease characterized by common symptoms such as fever, cough, and malaise, with severe cases potentially leading to chest tightness and hypoxemia. The pandemic has overwhelmed healthcare systems, disrupted essential services, highlighted inequalities, and impacted the physical and mental health of millions globally. According to the World Health Organization (WHO), as of January 31, 2024, over 1.3 billion cases and 10.2 million deaths have been confirmed worldwide. However, with the widespread administration of COVID-19 vaccines, most current infections result in mild symptoms. Consequently, most countries have relaxed stringent COVID-19 management measures, and on January 30, 2020, the WHO declared the end of the global emergency over COVID-19.

Corresponding authors
Yun Guo, guoyunsoochow@163.com
Tong Zhou, zhoutonghxk@163.com

Nonetheless, this does not imply that COVID-19 is entirely under control or can be disregarded. On the contrary, COVID-19 remains a global pandemic posing a substantial threat to public health, with infection rates continuing to rise (*World Health Organization (WHO), 2024b*). Additionally, a significant proportion of those infected experience post-COVID syndrome, which can affect nearly all organs and body systems, manifesting as fatigue, respiratory issues, neurological problems, cognitive impairments, and more (*Rochmawati, Iskandar & Kamilah, 2024*; *Morioka et al., 2023*; *Gallant et al., 2023*; *Salve et al., 2023*).

Cough is a common symptom in respiratory diseases and can be classified as acute (<3 weeks), subacute (3–8 weeks), or chronic (>8 weeks) based on its duration. Post-COVID cough is a significant feature of post-COVID syndrome, with a pooled prevalence ranging from 10% to 44% (*Rochmawati, Iskandar & Kamilah, 2024*), and has contributed to an increased number of outpatient visits. Although it is less likely to pose a severe threat to personal health, post-COVID cough can be particularly bothersome as it may occur unexpectedly, including during speech and sleep, leading to a reduced quality of life and disturbing others' rest. As the number of COVID-19 infections continues to rise, so does the number of patients experiencing post-COVID syndrome. Encouragingly, research on this population is also expanding. Predictors for the development of post-COVID syndrome include the severity of acute COVID-19 infection (*Jain et al., 2024*), smoking history, allergy history, COVID-19 vaccination doses (*Shang et al., 2024*; *Gentilotti et al., 2023*; *Fatima et al., 2023*; *Herman, Wong & Viwattanakulvanid, 2022*), and female sex (*Mahmoud et al., 2023*; *Gentilotti et al., 2023*; *Wan et al., 2023*). Gastrointestinal symptoms, sputum production, and a history of chronic cough prior to contracting COVID-19, rather than cough-specific quality of life, are significant predictors of cough-related outcomes in post-COVID-19 conditions (*Kanemitsu, Fukumitsu & Niimi, 2024*). However, there is currently limited research on the features and management of post-COVID cough. In this study, we included patients with subacute or chronic post-COVID-19 cough in the outpatient unit to analyze their characteristics, treatment, and outcomes.

## MATERIALS AND METHODS

This study was a single-center, observational, real-world study conducted in accordance with the STROBE guidelines (*von Elm et al., 2007*). The study flow is illustrated in Fig. 1. The study protocol was approved by the Ethics Committee of the Second Affiliated Hospital of Soochow University (JD-HG-2024017). Verbal informed consent was obtained from all patients or their families. The data were anonymized before statistical analysis and managed in accordance with standard data protection regulations.

### Study population

We collected data on patients admitted to our outpatient unit between August 2023 and February 2024 with a diagnosis of cough who met all the following criteria: (1) aged between 16 and 65 years; (2) cough following a documented COVID-19 infection,
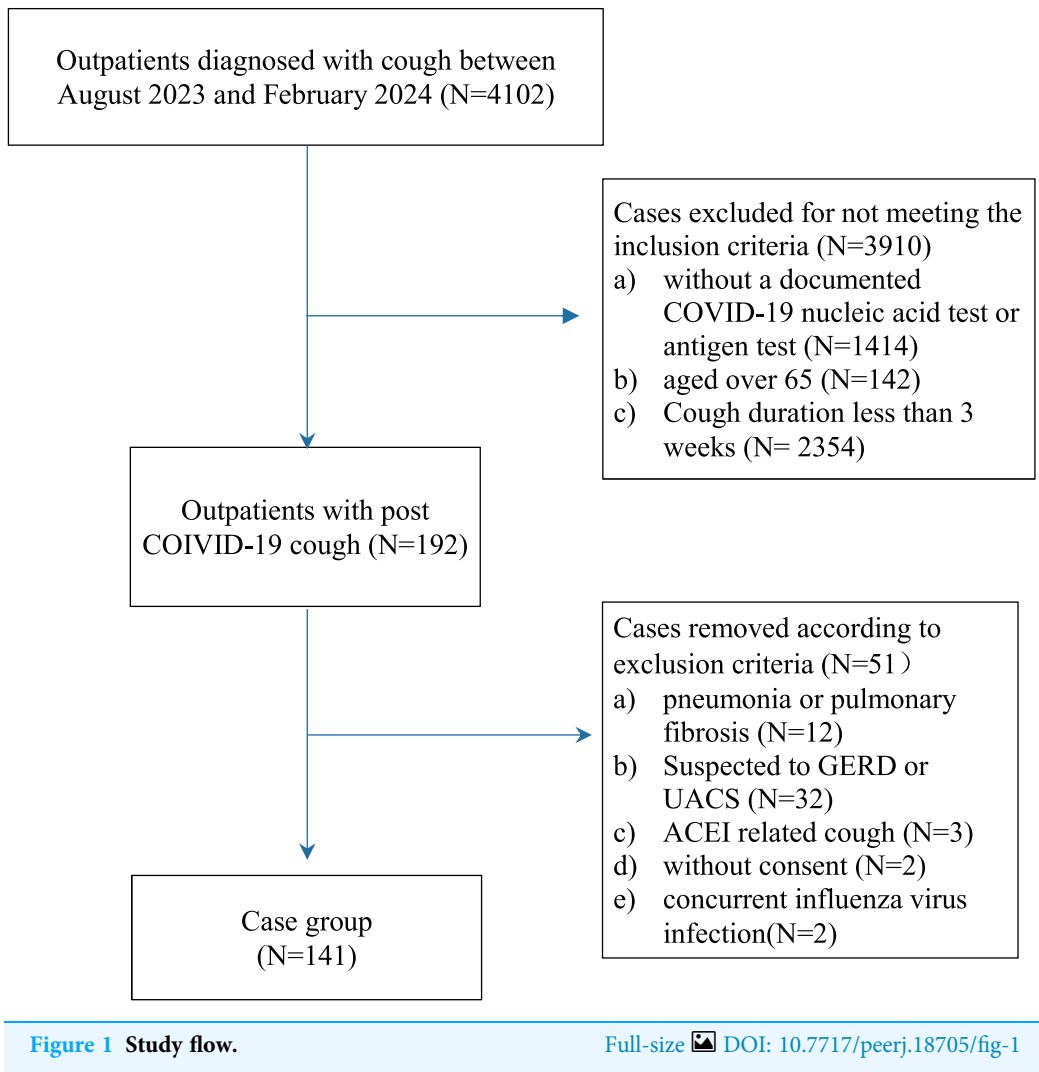

**Figure 1  Study flow.**                     

confirmed by a nucleic acid test or antigen test; (3) cough duration of at least 3 weeks; and (4) absence of fever.

To ensure that the cough could be attributed to a COVID-19 infection, patients with any of the following criteria were excluded: (1) recent radiographic evidence (X-ray or CT within 3 days) of pneumonia or pulmonary fibrosis; (2) suspected gastroesophageal reflux disease (GERD)-related cough (*e.g.*, acid reflux, heartburn) or upper airway cough syndrome (UACS) (*e.g.*, significant postnasal drip); (3) concurrent influenza virus infection, confirmed by a nucleic acid test or antigen test; (4) cough induced by angiotensin-converting enzyme inhibitors (ACEI), which resolved upon cessation of the drug; (5) lack of consent; or (6) asthma with poorly controlled symptoms.

## Clinical data collection and effectiveness assessment

The following data were collected: (1) demographic and baseline information, including age, gender, smoking history, vaccination status, comorbidities (such as hypertension and diabetes), and concurrent symptoms (*e.g.*, expectoration and abnormal laryngeal

sensations); (2) blood tests and equipment assessments, when available, including C-reactive protein (CRP), white blood cell (WBC) count, and fractional exhaled nitric oxide (FeNO) levels; and (3) prescription details and patient-reported cough severity during the initial visit and subsequent follow-up appointments.

Follow-up assessments were conducted *via* phone every 2 weeks following the patients' initial visit, with cough severity measured using the Visual Analog Scale (VAS), graded from 0 to 100. A VAS score of 70 indicated that the cough significantly impacted usual activities, such as sleep or talking. If a patient reported cessation of coughing during the first follow-up, the assessment was concluded; otherwise, a second follow-up was scheduled for 2 weeks later.

The efficacy of the treatment regimen was evaluated based on changes in VAS scores. A regimen was considered non-responsive (NR) if a patient's VAS score remained unchanged or increased. A complete response (CR) was defined as a decrease in VAS score by over 30 points or reaching a final VAS score of 0. Otherwise, the regimen was categorized as a partial response (PR). The final follow-up was conducted on March 7th, 2024.

## Statistical analysis

Continuous data were described as the mean ± standard deviation (SD) or median (interquartile range (IQR)) and were analyzed using t-tests or Wilcoxon signed-rank tests, depending on the data distribution. Categorical data were expressed as the number of cases and proportions, with the Chi-square test applied for hypothesis testing. Logistic regression analysis was employed to identify risk factors associated with cough relief (defined as a VAS decrease of 30 points or more). Variables with a $p$-value ≤ 0.05 in univariate logistic regression were further examined using multivariate logistic regression models. All statistical analyses and visualizations were performed using R (version 4.2.3; *R Core Team, 2023*) or GraphPad Prism (version 8.4.0). A $p$-value of < 0.05 was considered statistically significant.

## RESULTS

### Patient characteristics

A total of 4,102 patients were initially screened, of whom 192 were selected for further assessment. After excluding 51 patients, 141 individuals were ultimately included in the study.

As shown in Table 1, the patients had a median age of 42 years (mean ± SD: 42 ± 14), with females accounting for 70% of the cohort. The majority (117, 83%) had been vaccinated once (4, 3.4%) or more(113, 96.6%). A minority were smokers or ex-smokers (17, 12%).

The patients with subacute cough accounted for 68.1% of the population, and the median duration of cough was 8 weeks (interquartile range: 4–12 weeks). Dry cough was the only symptom reported by 85 patients (60%) while the other type was a cough with sputum, usually whitish. Additionally, over half of the patients (76, 54%) experienced abnormal laryngeal sensations, such as a dry or itchy throat. Diurnal variation in coughing

**Table 1 Patient characteristics.**

| Variable | Overall (N = 141) | Patients with medication (N = 131) | Patients without medication (N = 10) |
|---|---|---|---|
| Age | 42 (14) | 41 (14) | 43 (13) |
| Gender | | | |
| Female | 99/141 (70%) | 93/131 (71%) | 6/10 (60%) |
| Male | 42/141 (30%) | 38/131 (29%) | 4/10 (40%) |
| Cough duration (weeks) | 8 (4, 12) | 8 (4, 12) | 8 (6, 24) |
| Expectoration | 56/141 (40%) | 53/131 (40%) | 3/10 (30%) |
| Discomfort of throat | 76/141 (54%) | 71/131 (54%) | 5/10 (50%) |
| Smoking history | 17/141 (12%) | 14/131 (11%) | 3/10 (30%) |
| Hypertension | 10/141 (7.1%) | 9/131 (6.9%) | 1/10 (10%) |
| Diabetes | 4/141 (2.8%) | 3/131 (2.3%) | 1/10 (10%) |
| COVID vaccination | | | |
| Non-vaccinated | 24/141 (17%) | 24/131 (18%) | 0/10 (0%) |
| Vaccinated | 117/141 (83%) | 107/131 (82%) | 10/10 (100%) |
| WBC | | | |
| Normal | 17/141 (12%) | 16/131 (12%) | 1/10 (10%) |
| Not available | 124/141 (88%) | 115/131 (88%) | 9/10 (90%) |
| CRP | | | |
| Normal | 15/141 (11%) | 14/131 (11%) | 1/10 (10%) |
| Not available | 126/141 (89%) | 117/131 (89%) | 9/10 (90%) |
| FeNO | | | |
| Normal | 31/41 (76%) | 30/40 (75%) | 1/1 (100%) |
| High | 10/41 (24%) | 10/40 (25%) | 0/1 (0%) |
| Unknown | 100 | 91 | 9 |
| Cough time | | | |
| both | 45/141 (32%) | 44/131 (34%) | 1/10 (10%) |
| day | 29/141 (21%) | 25/131 (19%) | 4/10 (40%) |
| night | 67/141 (48%) | 62/131 (47%) | 5/10 (50%) |
| Primary VAS | 60 (50, 70) | 60 (50, 70) | 53 (50, 55) |

was insignificant for 32% of patients, while 48% experienced nocturnal coughing, and 21% coughed during the daytime.

Forty-one patients completed the fractional exhaled nitric oxide (FeNO) test, with 31 results within normal range and 10 cases (24%) exhibiting elevated levels. Routine blood tests were performed on 17 patients, and C-reactive protein (CRP) tests were conducted on 15 patients, both of which showed normal levels of white blood cells (WBC) and CRP.

## Treatment and outcome

Among the patients, 131 were prescribed oral medications. Compound methoxyphenamine (CMP) capsules were the most frequently prescribed, followed by inhaled corticosteroids/long-acting beta-2 agonist (ICS/LABA), montelukast sodium tablets, and Chinese patent drugs (CPD), including Suhuang Zhike capsules and Compound Loquat Dew. Ten patients

opted not to receive any medications based on their personal preference, given their normal auxiliary test results.

Throughout the follow-up period, patients who received prescribed medications demonstrated significant improvement. The median initial Visual Analog Scale (VAS) score for this group was 60 (interquartile range: 50–70), which decreased to 30 (interquartile range: 10–50) at 2 weeks, and further to 20 (interquartile range: 0–40) at 4 weeks. The overall response rate (ORR), defined as the sum of the complete response rate and partial response rate, was 86%. Among patients who opted not to take medications, the median initial VAS score was 53 (interquartile range: 50–55), decreasing to 30 (interquartile range: 30–40) at 2 weeks, and 30 (interquartile range: 21–38) at 4 weeks (as shown in Fig. 2). The natural overall response rate for this group was 80%. Patients who received prescribed medications showed greater improvement in VAS scores compared to those who did not, both at the 2-week time point ($26.5 \pm 20.8$ *vs.* $15.5 \pm 15.1$, $p = 0.054$) and at 4 weeks ($36.9 \pm 24.4$ *vs.* $21.0 \pm 18.4$, $p = 0.025$). We also compared the primary VAS scores of those who received vaccination ($n = 117$) with those who did not ($n = 24$), and the results indicated that those who received vaccination had a slightly lower primary VAS, but the difference was not significant ($56.41 \pm 16.77$ *vs.* $58 \pm 13.92$, $p = 0.47$).

To provide a more comprehensive analysis of the response to prescribed medications, patients were categorized into four groups based on the medication they were prescribed: the CMP group, montelukast group, ICS/LABA group, and CPD group. The CMP group had the largest number of patients ($n = 81$), with six patients exhibiting a partial response and 61 achieving a complete response. The ICS/LABA group, consisting of 46 patients, showed the highest complete response rate (CRR) of 87.0% ($n = 40$) and an overall response rate (ORR) of 95.7%. In the montelukast group, which included 46 patients, eight had a partial response and 30 had a complete response. The CPD group comprised 30 patients, of whom eight achieved a partial response and 16 a complete response (as shown in Fig. 3).

Logistic regression analysis was performed to identify factors associated with cough relief. The results indicated that patients with a cough duration of no more than 8 weeks (OR = 3.73, 95% CI [1.75–7.97], $p < 0.001$), those treated with ICS/LABA (OR = 4.44, 95% CI [1.72–11.51], $p = 0.002$), or those not treated with CPD (OR = 0.42, 95% CI [0.18–0.97], $p = 0.043$) were more likely to experience cough relief at 4 weeks. Multivariate logistic regression analysis, adjusted for age and gender, further confirmed that a cough duration of no more than 8 weeks (OR = 4.28, 95% CI [1.90–10.0], $p < 0.001$) and treatment with ICS/LABA (OR = 4.42, 95% CI [1.70–13.3], $p = 0.004$) were independently associated with cough relief (as shown in Fig. 4).

To elucidate the effectiveness of specific regimens, patients were categorized based on their detailed prescriptions. The detailed results are shown in Table 2. We listed the top six most frequent regimens for clinical reference. The regimen comprising CMP Capsules and montelukast was the most common, involving 27 patients, with an ORR of 70.4% and a CRR of 77.8%. This was followed by the regimen of CMP Capsules alone, which involved 26 patients and demonstrated an ORR of 76.9% and a CRR of 84.6%. The regimen containing only ICS/LABA or CPD was the third most frequent, involving 19 patients,

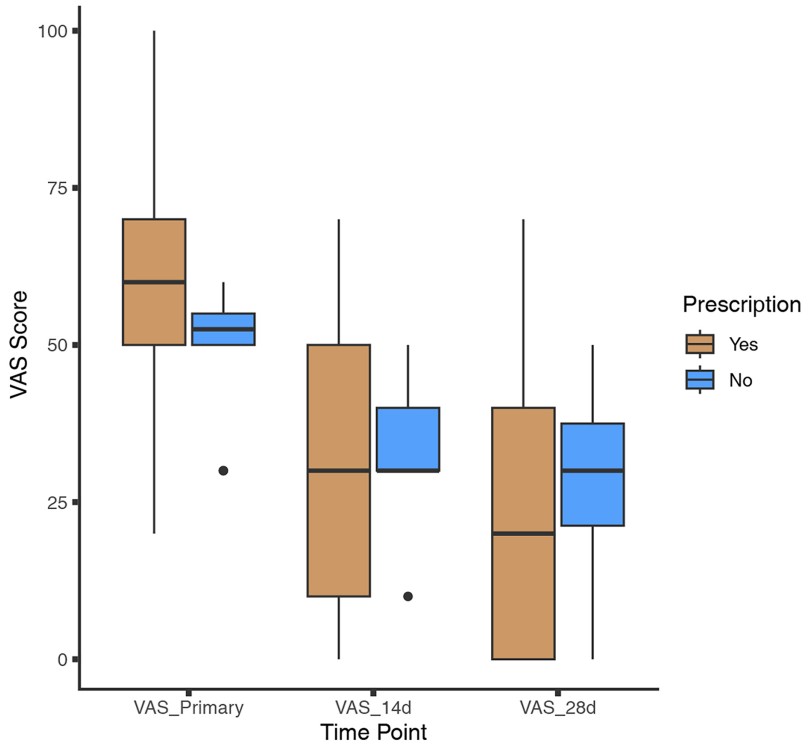

**Figure 2 Change in VAS score.** Patients were categorized based on whether they received medication. Visual analog scale (VAS) scores were collected at both the 2-week and 4-week time points.

with ORRs of 100% and 78.9%, and CRRs of 89.5% and 47.4%, respectively. Additionally, the regimens comprising CMP Capsules and ICS/LABA, and CMP Capsules combined with ICS/LABA and montelukast, involved 12 and nine patients, respectively. The ORRs and CRRs were as follows: for CMP Capsules + ICS/LABA, 83.3% and 83.3%, and for CMP Capsules + ICS/LABA + montelukast, 100% and 100%.

## DISCUSSION

In clinical practice, it is common for outpatients to face difficulties in completing all the necessary tests due to time constraints or financial limitations. However, it remains crucial for clinicians to take timely actions to alleviate their suffering. Coughing is one of the most distinct and recognizable symptoms of COVID-19 (*Pahar et al., 2022*), and advancements in machine learning have enabled the development of a COVID-19 cough classifier that can differentiate between COVID-19 positive and negative coughs (*Pahar et al., 2021*). Even when the viral load is well controlled, many patients experience a persistent, troublesome cough. Our study focused on patients suffering from subacute or chronic post-COVID cough in a real-world clinical setting. By summarizing their characteristics and treatment responses, we aim to provide valuable insights that can assist clinicians in managing this challenging condition.

We observed a predominance of females, which is consistent with findings related to other post-COVID-19 syndromes. Seventy-nine percent of patients reported coughing

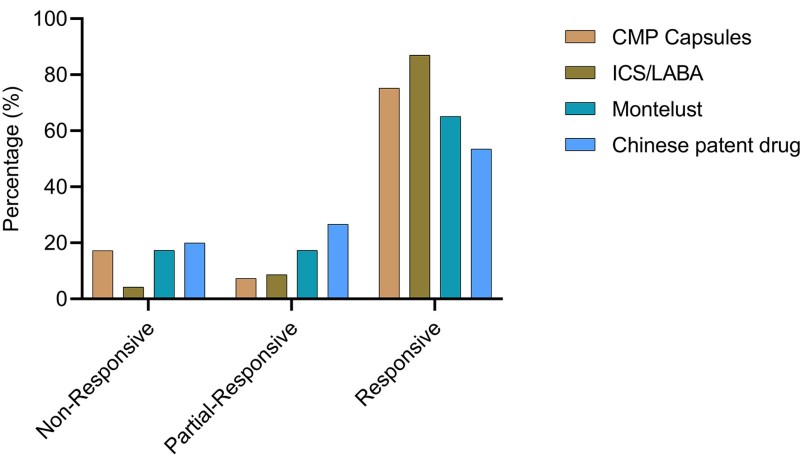

**Figure 3 Prescription and effectiveness.** Patients were classified into four groups (CMP, montelukast, ICS/LABA, and CPD) based on the corresponding prescribed medication. The effectiveness of each treatment was subsequently assessed.

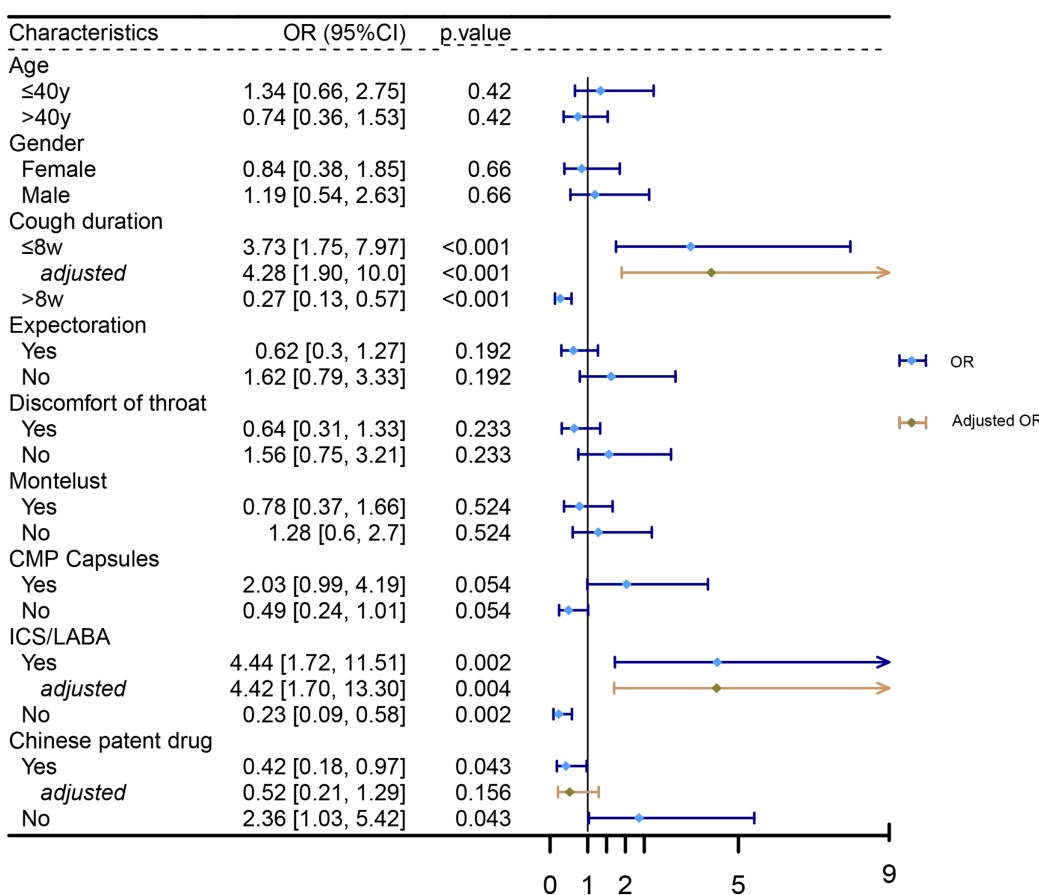

**Figure 4 Factors associated with cough relief.** Logistic regression analysis was performed to identify factors associated with cough relief.

**Table 2 Prescription and effectiveness.**

| Drug regimens | Samples | Non-response | Partial response | Complete response |
|---|---|---|---|---|
| CMP capsules | 26 | 4 (15.4%) | 2 (7.7%) | 20 (76.9%) |
| ICS/LABA | 19 | 0 (0.0%) | 2 (10.5%) | 17 (89.5%) |
| Chinese patent drug | 19 | 4 (21.1%) | 6 (31.6%) | 9 (47.3%) |
| Montelust | 6 | 0 (0.0%) | 4 (66.7%) | 2 (33.3%) |
| CMP Capsules + Montelust | 27 | 6 (22.2%) | 2 (7.4%) | 19 (70.4%) |
| CMP Capsules + ICS/LABA | 12 | 2 (16.7%) | 0 (0.0%) | 10 (83.3%) |
| CMP Capsules + Chinese patent drug | 5 | 0 (0.0%) | 2 (40.0%) | 3 (60.0%) |
| ICS/LABA + Montelust | 2 | 0 (0.0%) | 2 (100.0%) | 0 (0.0%) |
| ICS/LABA + Chinese patent drug | 4 | 0 (0.0%) | 0 (0.0%) | 4 (100%) |
| CMP Capsules + ICS/LABA + Montelust | 9 | 0 (0.0%) | 0 (0.0%) | 9 (100%) |
| CMP Capsules + Montelust + Chinese patent drug | 2 | 2 (100.0%) | 0 (0.0%) | 0 (0.0%) |

either at night or throughout the day, and most experiencing a dry cough. Additionally, over half of the patients reported abnormal laryngeal sensations. It was noted that while cough symptoms can gradually alleviate over time, the efficacy of medications tends to diminish more rapidly. We also collected data on prescriptions and recorded the responses to these regimens.

CMP capsules, which contain methoxyphenamine, aminophylline, chlorphenamine, and noscapine, were effective in improving symptoms of post-infectious cough (*Zhou et al., 2011*). These capsules were recommended for treating subacute SARS-CoV-2-associated cough in an expert consensus (*Branch, 2023*). In our study, CMP capsules were the most frequently prescribed, either alone or in combination with other drugs, and demonstrated acceptable effectiveness.

Montelukast, a leukotriene receptor antagonist, reduces swelling, mucus production, and bronchospasm. It has been reported as an ineffective treatment for post-infectious cough in a double-blind, multicenter, randomized placebo-controlled trial (*Wang et al., 2014*), though it is recommended for treating allergic rhinitis and asthma, including cough variant asthma (*Global Initiative for Asthma, 2023*). Despite this, montelukast showed moderate effectiveness in our study, possibly due to a relatively high rate of positive bronchodilator reversibility and FeNO test results. In *Gencer et al.*'s *(2023)* study, an increase in FEV1 of ≥200 ml was observed in 40 out of 151 (26.5%) patients with chronic cough and dyspnea lasting over 8 weeks. Nearly one-fourth of the patients in our study had a positive fractional exhaled nitric oxide (FeNO) test, which may also explain the promising effect of ICS/LABA, the most effective drug in our study.

The detailed mechanism of post-COVID-19 cough remains to be further elucidated. Cough is a reflex activated by stimuli in peripheral sensory nerves, which are transmitted to the central nervous system *via* the vagus nerve. When cough is induced by low levels of exposure due to aberrant amplification of stimuli in the pathway, it is termed cough hypersensitivity (*Canning et al., 2014*; *Mazzone et al., 2022*; *Morice et al., 2020*). Similar to chronic cough, several studies have proposed that cough hypersensitivity, caused by

neuronal hypersensitivity following COVID-19 infection, plays an important role (*Mohamed et al., 2022*; *Song et al., 2021*). This view is supported by *García-Vicente et al. (2023)*, who found that laryngeal electromyography revealed pathology in the thyroarytenoid and cricothyroid muscles in 76.3% of patients with chronic post-COVID-19 cough.

Our study provides valuable insights for clinicians managing post-COVID-19 cough. Our findings indicate that women are more frequently affected by post-COVID-19 cough, and importantly, patients can benefit from appropriate medication. Specifically, our data demonstrate that regimens including ICS/LABA are significantly associated with better outcomes, offering practical guidance for physicians in managing these cases.

Nevertheless, several limitations should be noted. Firstly, our study was conducted at a single hospital, which may limit the generalizability of our findings due to a restricted patient source. Secondly, the small number of participants in the final analysis may reduce the robustness of some findings. Notably, some treatment regimens showed efficacy rates of either 100% or 0%, which may be less convincing due to insufficient sample sizes. These results should be interpreted with caution. Future studies with larger sample sizes and multi-center designs are necessary to validate our findings and further investigate the clinical implications of subacute and chronic cough in COVID-19 outpatients.

## CONCLUSION

Our study indicates that women are more susceptible to developing a post-COVID-19 cough, and the majority of patients experienced relief after medication. Among the treatments evaluated, ICS/LABA was found to be the most effective. However, further prospective studies with larger samples are warranted to validate these findings.

### Funding

This work was supported by the Competitive discipline lift project of the Second Affiliated Hospital of Soochow University (grant number XKTJ-XK202007) and the Science and Technology Program of Suzhou (grant number SKY2021006). The funders had no role in study design, data collection and analysis, decision to publish, or preparation of the manuscript.

### Grant Disclosures

The following grant information was disclosed by the authors:
Second Affiliated Hospital of Soochow University: XKTJ-XK202007.
Science and Technology Program of Suzhou:  SKY2021006.

### Competing Interests

The authors declare that they have no competing interests.

## Author Contributions

- Chun Yao performed the experiments, analyzed the data, prepared figures and/or tables, and approved the final draft.
- Dongliang Cheng performed the experiments, analyzed the data, authored or reviewed drafts of the article, and approved the final draft.
- Wenhong Yang performed the experiments, analyzed the data, authored or reviewed drafts of the article, and approved the final draft.
- Yun Guo conceived and designed the experiments, authored or reviewed drafts of the article, and approved the final draft.
- Tong Zhou conceived and designed the experiments, authored or reviewed drafts of the article, and approved the final draft.

## Human Ethics

The following information was supplied relating to ethical approvals (*i.e.*, approving body and any reference numbers):

the Second Affiliated Hospital of Soochow University

## Data Availability

The raw measurements are available in the Supplemental File.

## Supplemental Information

Supplemental information for this article can be found online at http://dx.doi.org/10.7717/peerj.18705#supplemental-information.

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
