# Peer review of "Characterization and clinical outcomes of outpatients with subacute or chronic post COVID-19 cough: a real-world study"

_PeerJ, doi:10.7717/peerj.18705_

## Round 0.1 · original submission · Major Revisions

The data needs to be thoroughly analysed and methodologically experimented and validated.

The authors should consider the potential alternate causes of the coughing.

·

Basic reporting

Well written

Experimental design

The purppose was to describe post-Covid cough in patients at one center.

Validity of the findings

The description is valid as far as provided, but the presence absent when coughing or not when sleeping would be more relevant that just day or night coughing. If that information is available, I recommend it be substituted for night.

Additional comments

Unfortunately, their is no information about the cause of the cough. If availalble, the authors should describe in pneumonia was present (by radiology or clinically). For the patients with cough <8 weeks, that could be conistent with any post-viral cough. For those longer, the cause may be residual lung damage or habit cough which can begin after any viral respiratory infection and is not associated with any Covid induced lung damage. The treatments are interesting but should be acknowledged to possibly be the natural course without a placebo or no Rx.

·

Basic reporting

The quality of English used is clear, but minor improvements are needed.

The list of references can certainly be improved, as there are many recent articles on COVID-19 coughs.

The structure looks okay but needs some improvements as well, as explained later.

Experimental design

The paper only presents a simple statistical approach, which considers mean and standard deviation. It lacks the novelty in terms of data and methods. Considering medical data is hard to gather, 126 subjects is a good number, but it would be much nicer if there were more data. 
Considering this decent size of data, it would be good to carry out classification or regression analysis as well. This will enhance the novelties in the paper. 
Please use a figure to explain the tables, if possible. 

Finally, please summarise the novel findings.

Validity of the findings

This section is missing from the paper. The results are obtained using only a statistical significance test, where there is no way of carrying out validations. Running a classifier such as LR might help in validating the results.

Additional comments

This is a decent paper but requires improving in terms of novelties and methodologies. Unfortunately, I can't recommend it to be published at this stage, rather suggest a major revision.

---

## Round 0.2 · Major Revisions

As you can see, both reviewers still have concerns about your analysis. It is important that you address all these concerns, and satisfy the reviewers.

·

Basic reporting

Clear literature throughout; professional English used throughout.

Sufficient references.

Good article structure.

Hypothesis followed by supportive relevant data.

Experimental design

A limitation of the experimental design is the lumping of different patterns of cough, subacute and chronic, nocturnal and daytime, dry or wet. The relation to treatment may relate to pattern of cough and spontaneous remission is not distinguishable from medication response.

Validity of the findings

The relation to treatment may relate to pattern of cough and spontaneous remission is not distinguishable from medication response. That limits the potential for valid interpretation of treatment.

The most relevant and potentially meaningful conclusion is the different durations and types of cough. The data presentation should be limited to the detailed description of duration, wet or dry, day or night,etc.

Additional comments

The details of treatments provide no meaningful information. Limitation of the presentation to the details of the type and pattern of cough justify publication.

·

Basic reporting

The authors have improved the paper and it is easier to read now.

The list of references is also enlarged and explained in the paper along with considering larger and more recent COVID-19 data.

I am happy that all the conditions are met in this section.

Experimental design

Previously, the paper lacked many features but it has been improved on the revision.

I still strongly recommend that the authors use a classification or regression analysis to strengthen the paper even further. This is the one request I have in the second revision.

Validity of the findings

They are all good. This study is beneficial to many as the novel dataset has been well-explained.

The structure of the paper has also been improved.

Additional comments

I would recommend a revision where the authors either carry out classification/regression tasks or provide reasons for not carrying it out as the data presented in this paper is a novel one.

---

## Round 0.3 · accepted · Accept

Thank you for your detailed revision. I can confirm that all reviewers' comments have been fully addressed, and I am pleased with the current version. I believe it is now ready for publication.

·

Basic reporting

The authors have worked on the manuscript and it reads better than the previous version.

Experimental design

Thanks for working on my advice to implement a classifier, it has certainly improved the experimental design.

Validity of the findings

Logistic regression has enhanced the validity of findings, thanks for adding this.

Additional comments

N/A